# Preparation of W/O Hypaphorine–Chitosan Nanoparticles and Its Application on Promoting Chronic Wound Healing via Alleviating Inflammation Block

**DOI:** 10.3390/nano11112830

**Published:** 2021-10-25

**Authors:** Mengting Qi, Xuerui Zhu, Xiaoyi Yu, Min Ai, Weiwei Cai, Bin Du, Bao Hou, Liying Qiu

**Affiliations:** 1Department of Basic Medicine, Wuxi School of Medicine, Jiangnan University, Wuxi 214122, China; 6191502007@stu.jiangnan.edu.cn (M.Q.); 6201507020@stu.jiangnan.edu.cn (X.Z.); 6202805012@stu.jiangnan.edu.cn (X.Y.); 6182806001@stu.jiangnan.edu.cn (M.A.); caiweiwei@jiangnan.edu.cn (W.C.); dubin@jiangnan.edu.cn (B.D.); 2College of Pharmacy, Jiangnan University, Wuxi 214122, China

**Keywords:** HYP-NPS, chronic wound, IL-1β, TNF-α, inflammation

## Abstract

Chronic wound repair is a common complication in patients with diabetes mellitus, which causes a heavy burden on social medical resources and the economy. Hypaphorine (HYP) has good anti-inflammatory effect, and chitosan (CS) is used in the treatment of wounds because of its good antibacterial effect. The purpose of this research was to investigate the role and mechanism of HYP-nano-microspheres in the treatment of wounds for diabetic rats. The morphology of HYP-NPS was observed by transmission electron microscopy (TEM). RAW 264.7 macrophages were used to assess the bio-compatibility of HYP-NPS. A full-thickness dermal wound in a diabetic rat model was performed to evaluate the wound healing function of HYP-NPS. The results revealed that HYP-NPS nanoparticles were spherical with an average diameter of approximately 50 nm. The cell experiments hinted that HYP-NPS had the potential as a trauma material. The wound test in diabetic rats indicated that HYP-NPS fostered the healing of chronic wounds. The mechanism was through down-regulating the expression of pro-inflammatory cytokines IL-1β and TNF-α in the skin of the wound, and accelerating the transition of chronic wound from inflammation to tissue regeneration. These results indicate that HYP-NPS has a good application prospect in the treatment of chronic wounds.

## 1. Introduction

Chronic wounds not only bring distress to a patient’s life, but also bring economic pressure to their family [1,2]. From a demographic point of view, the number of patients with chronic wounds and impaired healing conditions is reaching epidemic proportions and will become heavier in terms of human health and the economy [3,4]. At present, there are 6.5 million patients with chronic incurable wounds caused by various reasons in the United States alone, and the annual investment in chronic wound treatment has reached approximately 20 billion dollars [5,6]. Therefore, more and more researchers are paying attention to the research of chronic trauma. The process of wound development can be summarized as hemostasis at the wound site, then entering the inflammatory infiltration stage, followed by the tissue regeneration stage, and finally the tissue repair stage [7,8]. In the inflammatory phase, the early inflammatory response mobilizes local inflammatory factors and the systemic defense response to the wound site. The inflammation period of chronic wounds lasts for a long time. Researchers believe that these wounds may be in a chronic inflammation state and fail transit to the tissue regeneration stage [9,10,11]. Studies have shown that pro-inflammatory cells composed of neutrophils and macrophages infiltrate the wound surface, leading to an increase in IL-1 β and TNF-α expression [12], which further leads to an increase in metalloproteinase and excessive degradation of local ECM, destroying cell migration and causing serious damage to host tissue [13,14].

Hypaphorine (HYP) is a small-molecule indole alkaloid in *Erythrina velutina* (Figure 1A). It has been reported that HYP from various marine sources has anti-inflammatory effects [15]. Previous studies have shown that HYP attenuates lipopolysaccharide-induced inflammation in RAW 264.7 cells and the endothelium [16,17]. HYP contains a carboxyl group, easy for hydroxy esterification, which makes it unstable and results in low bioavailability. In addition, the low molecular weight of HYP and its easy degradation limit its application.

Chitosan (CS), as a kind of polysaccharide, has been widely used because of its good biocompatibility and security [18,19,20]. In particular, CS is of wide interest because of its inherent advantages in promoting wound healing [21]. Moreover, according to its drug properties, CS can also be prepared into nanomaterials with drug loading function [22]. The prepared nanomaterials not only have good biocompatibility, but also have a sustained-release function. Hence, they can enhance the application range of drugs with different molecular structures [23].

Previous studies have confirmed the significant role of HYP and CS [15,21]. Hence, the design of this research aimed to enhance the basic performance of HYP by combining HYP and CS into W/O-type nanoparticles by the solvent diffusion–ionic crosslinking method. Then, the nanoparticles were applied to diabetic wound repair, exerting a local anti-inflammatory effect through the slow release of drug HYP at chronic wound sites, with CS inhibiting microbial reproduction at the wound site through its own antibacterial effect. Furthermore, the molecular mechanism of nanoparticles promoting wound healing are explored.

## 2. Methods

### 2.1. Drugs and Chemicals

Liquid paraffin and Span80 (Sinopharm Chemical Reagent Co., Ltd., Beijing, China). The grade of chitosan (C6H11NO4)n was food grade, its viscosity was less than 500 mpa·s (Solarbio Science & Technology Co., Ltd., Beijing, China). Hypaphorine (Shifeng Technology Co., Ltd., Shanghai, China). HuaTuo *Lithospermum* burn ointment (HT) (Yichang Kangmintang Pharmaceutical Co., Ltd., Shantou, China).

### 2.2. Determination of HYP

The content of HYP was determined by HPLC (waters 2695) with a C18 column (4.6 × 250 mm, 5 μm) and a UV detector at 280 nm. The mobiles were methanol (A) and 0.3% phosphoric acid solution (B) with a flow rate of 0.5 mL/min. The separation conditions were 0–10 min 35% (A)/65% (B), 10–20 min 35–40% (A)/65–60% (B), 20–35 min 40–50% (A)/60–50% (B).

### 2.3. Preparation of HYP-NPS Nanoparticles and Hydrogels

The nanoparticles were prepared by the solvent diffusion–ionic crosslinking method [24], in which 4 mL of Span-80 was added to 36 mL of liquid paraffin, mixed well in a 55 °C water bath, and then used as an oil phase. Then, 10 mg of HYP was dissolved in 8 mL of chitosan solution (5 mg/mL) and stirred evenly. The mixed solution of HYP and CS was slowly added to the oil phase and stirred at 680 r/min. After stirring in a water bath for 3 h, 2 mL of TPP solution (2 mg/mL) was slowly added and stirred for 0.5 h to obtain a uniform oily solution, which was washed thrice with 3 times the volume of petroleum ether, centrifuged, and vacuum-dried to obtain HYP nanoparticles. The HYP-NPS hydrogels were prepared by dissolving nanoparticles in an aqueous solution and ultrasonic mixing.

### 2.4. Characterization of HYP-NPS Nanoparticles

The sample was freeze-dried and then dissolved in ultrapure water. Then, 10 μL of the solution was added to a carbon-coated copper meshwork and stained with phosphotungstic acid, before being dried at 37 °C. TEM was used to observe the micromorphology of the HYP nanoparticles, the model of TEM was JEOL JEM-2100 (Tokyo, Japan), and the accelerating voltage was set to 120 kV.

The average nanoparticle size, zeta potential, viscosity, loading capacity, FT-IR spectra, and encapsulation efficiency (EE) of HYP-NPS were analyzed according to previous research methods [25]. Briefly, ZetaPALS was used to analyze the mean nanoparticle size and the zeta potential of 0.5 mg/mL of HYP-NPS solution. The model of ZetaPALS was Brookhaven Corporation. Each value was averaged from five parallel measurements. A rotary rheometer (DHR3; TA instruments, New Castle, DE, USA) was used to measure the viscosity of 5 mg/mL of HYP-NPS solution at room temperature (25 °C) and body temperature (37 °C). The shear rate was logarithmically distributed, ranging from 10^−1^ to 1–10^2^ s^−1^ (*n* = 3). FT-IR spectroscopy of HYP, NPS, and HYP-NPS was conducted with a potassium bromide precipitate on a spectrometer (Bruker Corporation, FT-IR TENSORII, Berlin, Germany) with a wave number range of 500–4000 cm^−1^ at a resolution of 4 cm^−1^.

### 2.5. Drug Release Assay In Vitro

First, 5 mg/m: HYP-NPS solution was prepared and release testing was performed after ultrasonic for 10 min. The, 2 mL was taken out and put it into the dialysis bag, which was then placed into the beaker of 100 mL of normal saline. The temperature of the receiving solution normal saline was controlled at 36–38 °C, and the magnetic stirrer was stirred at a uniform speed in the receiving tank. Next, 2 mL of normal saline was taken out for test at 0.5, 1, 2, 4, 6, 8, 10, 12, 24, and 36 h respectively. In addition, 2 mL of fresh normal saline was added at the same temperature. The normal saline of the receiving solution was filtered with a 0.22 μm filter membrane to remove impurities, and then 10 μL of filtrate was absorbed for detection. The concentration of hydroxyproline in normal saline of the receiving solution was measured. The cumulative release of HYP was calculated, and the relationship curve between cumulative release and time was drawn.

### 2.6. Cell Cytotoxicity Test

RAW 264.7 macrophages were used to detect the cytocompatibility of HYP-NPS. Briefly, RAW 246.7 cells were adjusted to 1 × 10^4^ cells/well, and 100 μL of RAW 246.7 cells per well were inoculated on a culture plate and cultivated in a standard environment. Then, the drugs were added to the wells. After the drug was added, the culture continued for 24 h, and every well was incubated for 4 h after adding 10 µL of the MTT solution with a concentration of 5 mg/mL. Then, 150 µL of dimethyl sulfoxide (DMSO) was added. The optical density (OD) value was determined at 570 nm. The following formula was used to calculate cell viability: Cell viability (%) = OD570 nm in cells treated with HYP-NPS hydrogel/OD.

### 2.7. Real-Time Quantitative PCR Analysis

The quantitative polymerase chain reaction analysis was performed to detect tumor necrosis factor-α (TNF-α) and interleukin-1β (IL-1β) mRNA expression. Trizol Reagent (Cwbio, Nanjing, China) was used to extract the total RNA in the chronic wound skin tissues, and 1 µg of total RNA was used for synthesis of the cDNA with Hifair III 1st Strand cDNA Synthesis SuperMix for Qpcr (Yeasen, Beijing, China). Afterward, the real-time quantitative PCR was executed with cDNAs and gene-specific primer pairs with Hieff UNICON Power qPCR SYBR Green Master Mix (Yeasen, Beijing, China) via a fluorescence quantitative LightCycler 480 Real-Time PCR system (Roche, Basel, Switzerland). Relative gene expression was calculated by the 2-△△CT method and β-Actin was used as reference for normalization.

The primers for TNF-α: 5′-CATCTTCTCAAAATTCGAGTGACAA-3′ (forward) and 5′-TGGGAGTAGACAAGGTACAACCC-3′ (reverse).

The primers for 1L-1β: 5′-GCTGAAAGCTCTCCACCTCAATG-3′ (forward) and 5′-TGTCGTTGCTTGGTTCTCCTTG-3′ (reverse).

The primers for β-Actin: 5′-CATGGAGTCCTGTGGCATCC-3′ (forward) and 5′-CTCCTTCTGCATCCTGTCGG-3′ (reverse).

### 2.8. Animal Experiment

All animals used in this research were provided by Shanghai Slake Experimental Animal Co., Ltd. (SCXK2017–0005) and were adaptively fed for seven days. Diabetic rats were induced by intraperitoneal injection of streptozotocin (STZ). STZ was dissolved in citrate buffer (10 mM, pH 4.5) and injected intraperitoneally at a dose of 70 mg/kg body weight [26]. After two weeks of STZ treatment, when the blood glucose was higher than 300 mg/dL, the rats were regarded as diabetic, and then the open wound model was created. All animals are kept in a standard barrier environment. The feeding and experimental procedures were implemented in accordance with the regulations of the experimental animal center of the Medical College of Jiangnan University. The ethical approval code of this research is JN.No20201030c0750315.

### 2.9. In Vivo Wound Healing Study

All rats (48 male SD rats) were divided between a control group (only smeared with normal saline for wound treatment; CON), a HYP drug group (HYP saline solution smeared for wound treatment), a nanoparticle hydrogel control group (for wound smear without loading nanohydrogel; NPS), a nanoparticle-loaded HYP drug group (HYP-NPS), and a HT-positive drug control group (HT).

*Lithospermum officinale* is an herbal species of the genus *Lithospermum*. It has clinical efficacy in inhibiting inflammation of skin diseases [27,28]. HuaTuo *Lithospermum* burn ointment (HT) has anti-inflammatory and antibacterial effects, and has been widely used in the treatment of trauma. The main components of HuaTuo *Lithospermum* burn ointment are provided in the Appendix A.

The diameter of the wound was measured with a scale every day and pictures were taken with a camera. On the 6th, 9th, and 12th days, the skin of the wound site was taken for immunohistochemistry and Western blot analyses. The healing rate of the chronic wounds on the backs of rats was calculated according to the following method: Healing rate (%) = [(area on day 0 − open area on day n)/area on day 0] × 100.

### 2.10. Histopathology Study

The skin tissue was separated into 5 μm tissue slices. Then, hematoxylin eosin (H&E) staining was performed for pathological research. The pathological status of the wound skin was analyzed under a microscope. On the 6th, 9th, and 12th days after treatment, four rats in each group were euthanized with isoflurane. Afterward, the chronic wound skin tissue was collected and fixed immediately in 10% neutral buffered formalin (PH. 7.12) for one week. Then, the skin tissue was rinsed with running water, embedded in paraffin, and then separated into 5 micron-thick sections. Finally, the sections were stained with hematoxylin and eosin (H&E). The degree of epithelialization, the number of angiogenesis, and the formation of granulation tissue were compared in different groups.

### 2.11. Immunohistochemical (IHC) Staining

The immunohistochemical (IHC) staining steps were as previously described [25]. The antibody of IL-1β was diluted 100 times and added to the tissue section overnight. Finally, BCIP/NBT was employed for staining before imaging. Lastly, the sections were counterstained with the hematoxylin staining solution. Six regions were randomly selected from the tissue section for photographing, and IPP software was used to calculate the area of IL-1β-positive expression.

### 2.12. Western Blotting

Western blotting was used to analyze the protein expression level of the skin tissue in the chronic wounds. The Western blot sequences were as previously described [25]. Briefly, a BCA Protein Assay Kit (Beyotime, Nanjing, China) was used to quantify the protein concentrations of the rat skin tissue. The antibodies IL-1β (1:1000) and TNF-α (1:200) were diluted according to the requirements of the experiment manual. ImageJ software (Bio-Rad, Hercules, CA, USA) was used to semi-quantitatively analyze the final exposed protein band images.

### 2.13. Statistical Analysis

All results are expressed as mean ± SD from at least three independent experiments. Moreover, *t*-tests were used for comparisons between two groups. For multiple group comparisons, statistical analysis was performed by ANOVAs, followed by Dunnett’s tests. A *p*-value of <0.05 was taken as significant.

## 3. Results and Discussion

### 3.1. Physical–Chemical Characterization of HYP Nanoparticles (HYP-NPS)

The morphology of HYP-NPS was observed by TEM (Figure 1B). It can be seen from the image that the diameter of the nanoparticles was approximately 50 nm, and the nanoparticles had a spherical structure. In the process of sample freeze-drying, water evaporation shrank the nanoparticles and reduced their particle size. However, due to the existence of the oil film on the surface of the nanoparticles, the morphology of the nanoparticles did not change. The average particle size was approximately 64.97 ± 4.39 nm, as measured by ZetaPALS (Figure 1C). In addition, due to the existence of the oil film on the surface of the nanoparticles, the particles also formed partial aggregation. The zeta potential of HYP-NPS was approximately −1.83 ± 0.32 mV (Figure 1D).

Freeze high-speed centrifugation was used to separate the HYP-NPS solution. Then, HPLC was performed to analyze the LC and EE of HYP in the nanoparticles. The results showed that the LC and EE were 18.99 ± 1.41% and 89.07 ± 2.83%, respectively (Figure 1E,F). The nanoparticles prepared by the W/O method demonstrated good encapsulation efficiency for drug HYP.

As an excellent skin wound repair material, in addition to promoting wound healing, good adhesion is also essential [29]. The results showed that HYP-NPS hydrogels have good viscosity profiles between 10^−4^ and 10^−1^ Pa·s. The viscosity properties of the HYP-NPS hydrogels under different temperatures of 25 °C and 37 °C were obtained (Figure 1G,H).

### 3.2. FT-IR Analysis

A combination mode between NPS and HYP is crucial for the production of high-performance and good-efficacy HYP-NPS [30]. The combination mode between HYP and NPS were studied by Fourier transform infrared spectroscopy.

From Figure 2A, we can see that the absorption bands of hydroxyl and amino bonds were at 2800–3100 cm^−1^, and the stretching vibration carbonyl group was at 1630 cm^−1^. The peak at 1350 cm^−1^ was C–N vibrations of HYP. The absorption band at 900–600 cm^−1^ was caused by the out-of-plane bending vibration of C-H in aromatics.

The O–H and N–H tensile peaks of NPS were observed at 3290 cm^–1^. Obviously, the C=O stretching vibration peak was at 1558 cm^−1^ [31], and the peak of C=O symmetric stretching was at 1409 cm^−1^. Apparently, a strong peak of P–O for PO_4_ was observed at 1029 cm^−1^, and this strong peak may be caused by both COH stretching and C–N vibrations.

HYP-NPS had a lot of band types similar to NPS, but the peak value and intensity decreased. This confirms the interaction between HYP and NPS. Obviously, the peak in NPS at 3290 cm^−1^ shifted to 3309 cm^−1^, shown in the HYP-NPS spectrum, which indicates that the intramolecular/intermolecular hydrogen bond was reinforced. Furthermore, owing to the function of HYP, the characteristic peak of C=O at 1558 cm^−1^ and 1409 cm^−1^ moved to 1555 cm^−1^ and 1403 cm^−1^, respectively.

### 3.3. Sustained Release Test

Figure 2B shows the release curve of the HYP-NPS solution. The HYP release rate was 48.69 ± 0.22% in 0–12 h, and gradually slowed down after 12 h, arriving at 58.14 ± 0.33% in 36 h. The release efficiency in vitro was low because the nanoparticles were encapsulated by an oil film. In addition, the combination of nanoparticles and drug HYP through intermolecular force also led to the slow release of HYP. However, when the nanoparticles were adsorbed into the wound tissue, the oil film was combined with the tissue, and HYP was slowly released to the chronic wound site.

### 3.4. Cytocompatibility Assessment of HYP-NPS

Good cytocompatibility is an essential factor for biomaterials [31]. Cytocompatibility of the prepared HYP-NPS was assessed using the MTT assay kit at 24 h after RAW 264.7 cell seeding. As can be seen from Figure 3, HYP, NPS, and HYP-NPS showed cytocompatibility up to 24 h, but did not promote the proliferation of RAW 264.7 macrophages. The results showed that the RAW 264.7 cells had good viability, indicating the wonderful cytocompatibility of HYP-NPS. The above experiments confirm that HYP-NPS has broad application prospects.

### 3.5. Effect of HYP-NPS on LPS-Induced RAW 264.7 Responses In Vitro

LPS is a polysaccharide found in the cell wall of Gram-negative bacteria, which is responsible for activating the release of a variety of pro-inflammatory cytokines (IL-1β and TNF-α) in macrophages [32]. Due to the RAW 264.7 macrophages being stimulated by LPS for 24 h, the IL-1β (Figure 4A) and TNF-α (Figure 4B) mRNA expression level increased significantly. HYP, NPS, and HYP-NPS could effectively counteract the IL-1β and TNF-α mRNA levels in LPS-challenged RAW 264.7 cells. In particular, the mRNA levels of IL-1β and TNF-α induced by LPS in the HYP group were significantly lower than those in the NPS and HYP-NPS groups (Figure 4A,B). The inhibitory effect of the HYP-NPS group on inflammation was better than that of the NPS group. However, the inhibitory effect of the HYP-NPS group on inflammatory factors was not better than that of the HYP group, which may be due to the slow release of HYP caused by the encapsulation of HYP by nanoparticles, which could not directly act on macrophages. These results indicate that HYP and HYP-NPS had better effects of inhibiting chronic wound inflammation in vitro.

### 3.6. Effect of HYP-NPS on Wound Healing in Diabetic Rats

Diabetic wounds are a typical chronic wound. The main symptoms of a chronic wound include persistent inflammatory infiltration, prolonged infection, drug-resistant microorganisms, and epidermal cells that do not respond to repairing stimuli [33,34]. These pathophysiological phenomena lead to wound healing failure.

Wound healing was observed on the 0, 3rd, 6th, 9th, and 12th days (Figure 5A). Wound healing was faster and better in the HYP-NPS group than in the other groups. Figure 5B shows photographic pictures of the diabetic rats’ wounds in back skin administrated NPS, HYP, HYP-NPS, and positive drug HuaTuo *Lithospermum officinale* burn ointment. On the third day after wound treatment, the control group showed obvious characteristics of chronic wound nonunion, while the other groups had begun to scab. On the sixth day, the wound healing rate of the HYP-NPS group approached 70%, which was better than that of NPS and control group. On the ninth day, wound recovery was obvious in the HYP-NPS group, and the wound healing rate was superior to that of the HYP group, and there was no significant difference to the commercial drug HT group. With the delay of time, on the 12th day, the wound in the HYP-NPS administration group had healed. However the wounds in the control and NPS groups were still apparent. The above results indicate that HYP-NPS can accelerate the healing of chronic wounds. The wound healing rate of the HYP-NPS group was significantly faster than that of the HYP group, because the chitosan nanoparticles had a good antibacterial effect at the wound site in the HYP-NPS group. Therefore, HYP-NPS may be a potential candidate for the remedy of chronic wound in vivo.

### 3.7. Histopathological Results

In order to investigate the role of HYP-NPS on the histomorphology of the diabetic rats’ wound skin, HE staining was used to analyze the pathological change. Existing research data show that chronic wounds contain aging keratinocytes, endothelial cells, fibroblasts, and macrophages [35]. Vascular regeneration was slow [36], collagen deposition was delayed, and granulation tissue grew slowly [37].

HE staining of the skin of diabetic rats is shown in Figure 6. On the six postoperative day, the inflammatory cells in the skin tissue of the HYP-NPS group were significantly lower than those in the other groups, indicating that the inflammatory infiltration attenuated during the regeneration of traumatic tissue. In addition, the vascular regeneration at the wound site was obvious in the HT and HYP-NPS groups. Then, on the ninth day, the wound epithelial tissue began to form gradually. The re-epithelialization of the HYP-NPS and HT groups was significantly faster than that of the other groups. As can be seen from Figure 6B, collagen began to deposit gradually in the HYP-NPS group. The inflammatory cells in the control and NPS groups were obvious, which indicates that the wound skin was still in the state of inflammatory infiltration. On the 12th day after wound treatment, re-epithelialization was completed in the HYP-NPS and HT groups, and fibroblasts also began to appear in large numbers. Collagen deposition and vascular regeneration also began to appear in the other groups. These results indicate that HYP-NPS can promote the re-epithelialization of wound tissue by inhibiting the inflammatory reaction of chronic wounds.

### 3.8. Expression Analysis of Inflammatory Factors

IL-1β is secreted by a variety of inflammatory cells, which is the first signal to warn of damage to the surrounding cell barrier [38,39]. When the tissues and cells are damaged and stimulated, TNF-α is rapidly released by macrophages. TNF-α is also the most important cytokine secreted by various cells [40,41].

The results of Western blotting and immunohistochemistry analysis after the 6th, 9th, and 12th days of skin injury are presented in Figure 7. With the healing of the chronic wounds, the demonstration of IL-1β and TNF-α decreased gradually. The expression of IL-1β in the HYP-NPS group was considerably lower than that in the other groups; however, TNF-α did not change greatly by the sixth day after wound treatment. On the ninth day, with the healing of the chronic wounds, the expression of TNF-α in each group diminished slowly. The expression of IL-1β also began to decline, except in the control group. On the 12th day, immunohistochemistry and Western blotting examination showed low levels of IL-1β and TNF-α in the HYP-NPS group. All of the above results demonstrate that HYP-NPS plays a role in the inflammatory phase of chronic wounds, improves inflammatory block, and speeds up the transition from chronic wounds to tissue regeneration.

Substantial evidence suggests that in chronic ulcers, the infiltration of pro-inflammatory cells mainly composed of neutrophils and macrophages increases, leading to delayed wound healing [42,43]. Macrophages can release a variety of cytokines and growth factors in the process of the inflammation phase, and play the three main roles of antigen presentation, phagocytosis, and immune regulation [44]. Excessive secretion of pro-inflammatory cytokines by macrophages can result in damaging inflammation of new tissue in wound skin [45].

## 4. Conclusions

In the present study, W/O-type HYP-NPS were successfully prepared by the solvent diffusion–ionic crosslinking method as wound healing materials. HYP-NPS showed better physical and mechanical properties than a HYP solution, perhaps making it more appropriate for chronic wound healing. The encapsulation efficiency of HYP in NPS was 89.07 ± 2.83% and the loading capacity was 18.99 ± 1.41%. The diameter of the nanoparticles was approximately 50 nm, and the nanoparticles had a spherical structure. A cell viability assay showed adequate cytocompatibility of HYP-NPS. Animal assays demonstrated that HYP-NPS can hasten the wound healing rate of diabetic rats. HYP-NPS promoted the wound healing of diabetic rats was by down-regulating the expression of IL-1β and TNF-α in the wound site, accelerating the transition of chronic wounds from inflammation to tissue regeneration. In consideration of the properties of HYP-NPS and the nature of HYP drugs, together with their application effects, the commercialization of HYP-NPS for diabetic wound repair appears very useful.

## Figures and Tables

**Figure 1 nanomaterials-11-02830-f001:**
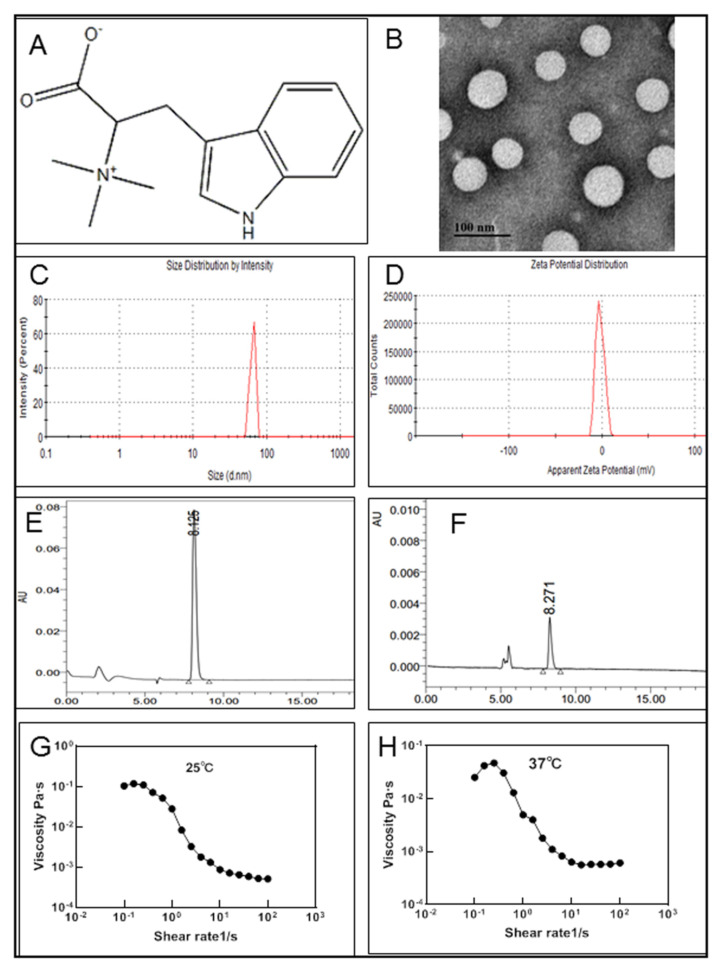
**Physicochemical characteristics of HYP-NPS**. (**A**). Molecular structure of hypaphorine. (**B**). TEM of HYP-NPS. (**C**,**D**). Particle size distribution data and zeta potential of HYP-NPS. (**E**) HPLC detection of hypaphorine. (**F**). HPLC detection after centrifugation. (**G**,**H**) Viscosity properties of HYP-NPS hydrogels under 25 °C (**G**) and 37 °C (**H**) (*n* = 3).

**Figure 2 nanomaterials-11-02830-f002:**
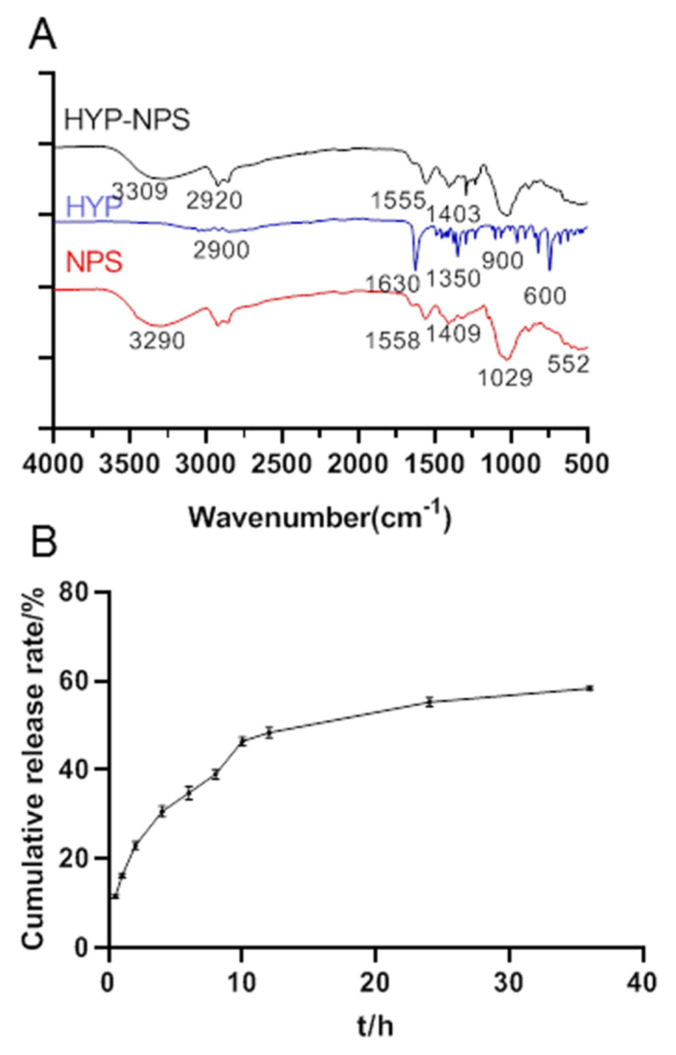
(**A**) The FT-IR spectra of HYP-NPS nanoparticles. (**B**) Sustained release curves of HYP from HYP-NPS solution (*n* = 3).

**Figure 3 nanomaterials-11-02830-f003:**
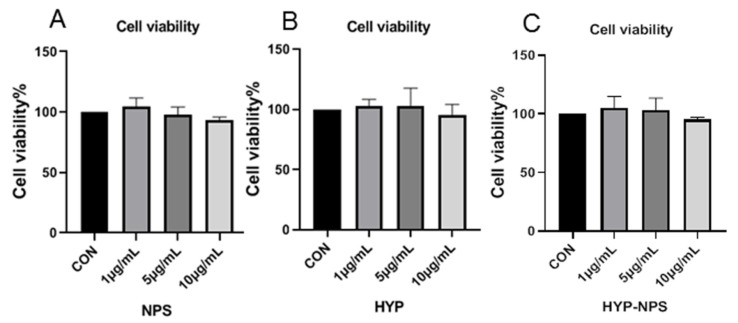
Biocompatibility assessment of NPS, HYP, and HYP-NPS.

**Figure 4 nanomaterials-11-02830-f004:**
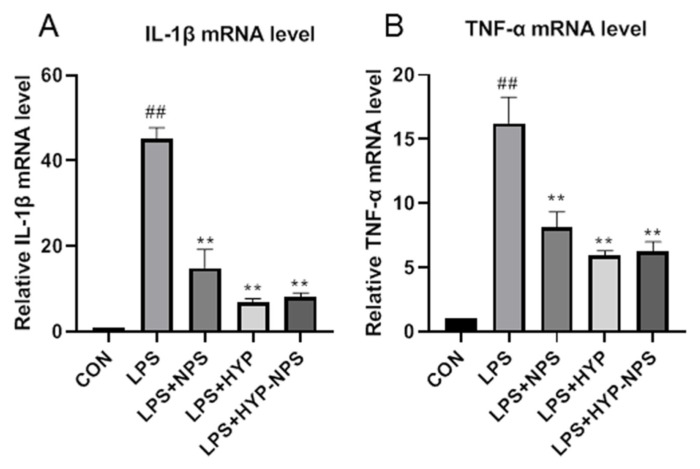
Effects of HYP (5 μg/mL), LPS + NPS (25 μg/mL), and LPS + NPS-HYP (25 μg/mL) on the mRNA expressions of IL-1β (**A**) and TNF-α (**B**), responding to LPS (1 μg/mL)-treated RAW264.7 cells for 24 h in vitro. Values are mean ± SD. ^##^
*p* < 0.01 vs. control, ** *p* < 0.01 vs. LPS. *n* = 6 for each group.

**Figure 5 nanomaterials-11-02830-f005:**
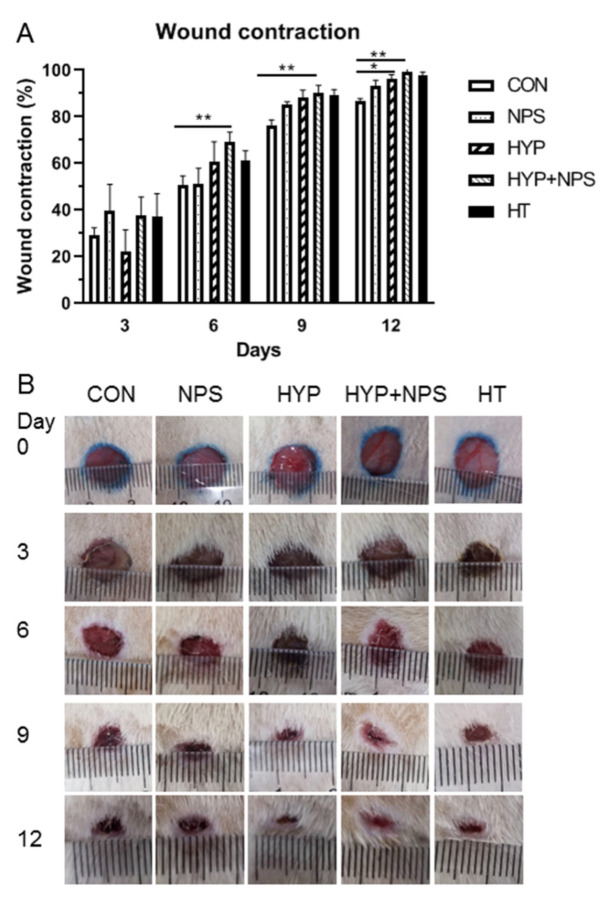
In vivo wound-healing results: (**A**) Histogram comparing the wound closure on the 3rd, 6th, 9th, and 12th days post-wounding. (**B**) Photographs of the macroscopic appearances of wounds treated on the 3rd, 6th, 9th, and 12th days post-wounding. * *p* < 0.05 and ** *p* < 0.01.

**Figure 6 nanomaterials-11-02830-f006:**
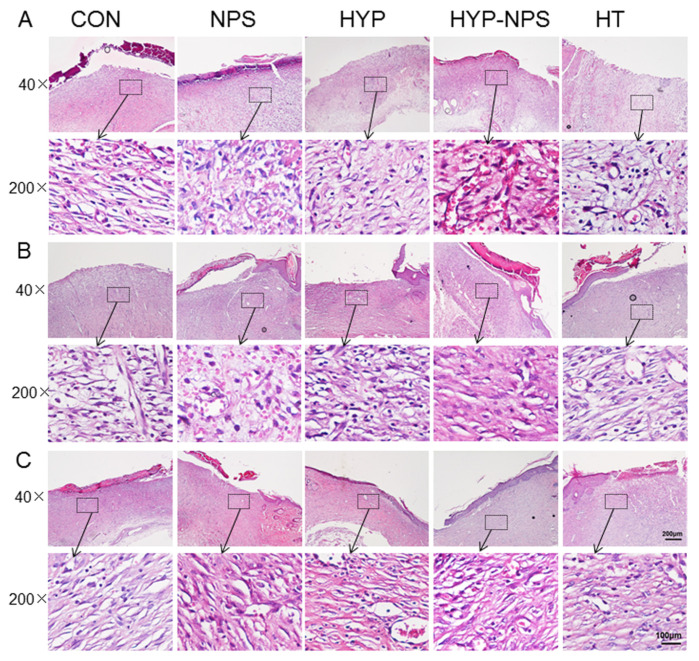
HYP-NPS inhibits the inflammatory response and accelerates re-epithelialization. H&E-stained microscopic sections of healed incisions in rats on the 6th (**A**), 9th (**B**), and 12th (**C**) days. I represent inflammatory cells, B represents blood vessels, and F represents fibroblasts.

**Figure 7 nanomaterials-11-02830-f007:**
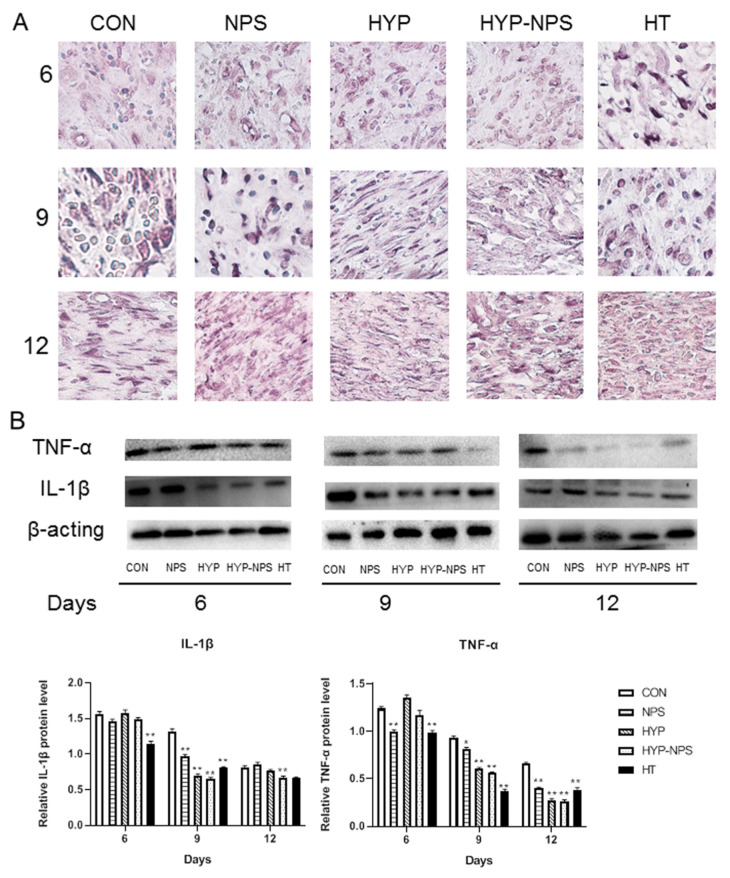
**HYP-NPS can down-regulate IL-1β and TNF-α expression levels.** (**A**) IHC staining of skin sections on the 6th, 9th, and 12ths day post-wounding (magnification 400×). (**B**) Western blotting showing protein expression levels of TNF-α and IL-1β. Values are mean ± SD. * *p* < 0.05 and ** *p* < 0.01 compared to the CON group. The arrows represent IL-1β expression.

## Data Availability

The data presented in this study are available on request from the corresponding author.

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
