# Peer review of "Preparation of W/O Hypaphorine–Chitosan Nanoparticles and Its Application on Promoting Chronic Wound Healing via Alleviating Inflammation Block"

_nanomaterials, 2021, doi:10.3390/nano11112830_

Round 1

Reviewer 1 Report

Tc

In the introduction, you should provide some examples of diseases which leads to chronic wound emphasizing its impact on society.

You mentioned that “Previous studies have confirmed the significant role of HYP and CS”. Could you please provide the references of those studies and major achievements regarding wound healing.

In the results, could you explain this sentence “The improved the preparation method of nanoparticles by W/O process, which greatly improved the encapsulation efficiency of HYP.”

Furthermore, you mention that HYP-NPS hydrogels showed good viscosity profiles. Define what you classification regarding its applications please.

Regarding the release of HYP, you mention that “The release efficiency in vitro was low because the nanoparticles were encapsulated by oil film”. You should discuss better this regarding HYP chemistry and nanoparticles combination with HYP revealed in FT-IR.

Regarding the in vitro inhibition of inflammatory cytokines, can you provide the time of LPS stimulation that you used. Your results do not demonstrate a clear improvement of using HYP encapsulated in NPs, however is evident that HYP-NPs compared with NPs alone is better. You should discuss better those results to provide a better understanding of the advantage of using HYP-NPs.  The same comment is valid for the in vivo results.

Regarding the conclusion, please rephrase the following sentence” Cell viability assay showed nontoxicity of HYP-NPS and had good biocompatibility” by showed adequate cytocompatibility since biocompatibility is not the correct term. Please also check the English in this section, which should be revised by a native speaker.

Author Response

We are very grateful to the reviewer for his/her valuable comments and suggestions on our manuscripts. In the manuscript, we marked the revised parts in red. The answers to the questions of editors and reviewers are as follows:

Q1: In the introduction, you should provide some examples of diseases which leads to chronic wound emphasizing its impact on society.

ANS: Thank you very much for your valuable advice. We added these contents in the introduction section. The specific modifications are as follows: Chronic wound will not only bring distress to the patient's life, but also bring eco-nomic pressure to the patient's family[1,2].From a demographic point of view, the number of patients with chronic wounds and impaired healing conditions is reaching epidemic proportions and will become heavier in terms of human health and economy[3,4].At present, there are 6.5 million patients with chronic incurable wounds caused by various reasons in the United States alone, and the annual investment in chronic wound treatment reaches about 20 billion dollars [5,6].Therefore, more and more researchers pay attention to the research of chronic trauma.

Q2: You mentioned that “Previous studies have confirmed the significant role of HYP and CS”. Could you please provide the references of those studies and major achievements regarding wound healing.

ANS: Thank you for your reminding. In the introduction of HYP, we introduced the anti-inflammatory effect of HYP, and in the introduction of chitosan, we introduced the role of chitosan in promoting wound healing and antibacterial. We attach references 15 and 21 to that paragraph.

Q3:In the results, could you explain this sentence “The improved the preparation method of nanoparticles by W/O process, which greatly improved the encapsulation efficiency of HYP.”

ANS: In fact, we have done a lot of work to realize the encapsulation of drug HYP by nanoparticles. The finally prepared nanoparticles not only have good encapsulation efficiency for drug HYP, but also have good sustained-release effect. However, due to the length of the article, we did not list this part of the work. Therefore, we change this sentence to that the nanoparticles prepared by W/O method have good encapsulation efficiency for drug HYP.

Q4: Furthermore, you mention that HYP-NPS hydrogels showed good viscosity profiles. Define what you classification regarding its applications please.

ANS: Thanks to the reviewer's reminder, we forgot to add references to this paragraph. As an excellent skin wound repair material, in addition to the good promoting effect of the ingredients on the wound repair process, good adhesion performance is also necessary. We refer to the Executive Standards in the following references. The chemistry and engineering of polymeric hydrogel adhesives for wound closure: a tutorial

Q5:Regarding the release of HYP, you mention that “The release efficiency in vitro was low because the nanoparticles were encapsulated by oil film”. You should discuss better this regarding HYP chemistry and nanoparticles combination with HYP revealed in FT-IR.

ANS: Thanks to the comments of review experts, we have made the following adjustments to the description of this sentence .The release efficiency in vitro was low because the nanoparticles were encapsulated by oil film. In addition, the combination of nanoparticles and drug HYP through intermolecular force will also lead to the slow release of HYP.

Q6:Regarding the in vitro inhibition of inflammatory cytokines, can you provide the time of LPS stimulation that you used. Your results do not demonstrate a clear improvement of using HYP encapsulated in NPs, however is evident that HYP-NPs compared with NPs alone is better. You should discuss better those results to provide a better understanding of the advantage of using HYP-NPs.  The same comment is valid for the in vivo results.

ANS: Thanks to the review experts' reminder. In this paper, we forgot the induction time of LPS on RAW264.7 cells. The mRNA expression levels of IL-1β and TNF-α were measured 24 h after LPS stimulation of 264.7 macrophages. We made the following modifications in the article.

LPS is a polysaccharide found in the cell wall of Gram-negative bacteria, which is responsible for activating the release of a variety of pro-inflammatory cytokines (IL-1β, TNF-α) in macrophages[29]. Due to the raw 264.7 macrophages are stimulated by LPS for 24 hours, resulting in IL-1β (Fig.4A), TNF-α (Fig.4B) mRNA expression level increased significantly. HYP, NPS and HYP-NPS could effectively counteracted IL-1β and TNF-α mRNA levels in LPS-challenged RAW 264.7 cells. In particular, the mRNA levels of IL-1β and TNF-α induced by LPS in HYP group were significantly lower than those in NPS group and HYP -NPS group (Fig.4A and 4B). The inhibitory effect of HYP - NPS group on inflammation was better than that of NPS group. However, the inhibitory effect of HYP-NPS group on inflammatory factors was not better than that of HYP group, which may be due to the slow release of HYP caused by the encapsulation of HYP by nanoparticles, which could not directly act on macrophages. These results indicated that HYP and HYP-NPS had better effect of inhibiting chronic wound inflammation in vitro.

In the discussion of the experimental results in vivo, we have also made corresponding modifications.

Q7:Regarding the conclusion, please rephrase the following sentence” Cell viability assay showed nontoxicity of HYP-NPS and had good biocompatibility” by showed adequate cytocompatibility since biocompatibility is not the correct term. Please also check the English in this section, which should be revised by a native speaker.

ANS: Thank you very much for your valuable advice. We have modified this part in the results and discussion. The specific modifications are as follows:

Good cytocompatibility is an essential factor for biomaterials [28]. Cytocompatibility of the prepared HYP-NPS was assessed using the MTT assay kit at 24 hours after RAW264.7 cells seeding. As can be seen from Figure 3, HYP, NPS and HYP-NPS showed cytocompatibility up to 24 h, but did not promote proliferation of RAW264.7 macrophages. The results showed that RAW264.7 cells had good viabilities, indicating the wonderful cytocompatibility of HYP-NPS. The above experiments confirm that HYP-NPS has broad application prospects.

Reviewer 2 Report

The manuscript shows a complete work, which explore, for the first time the preparation of HYP- NPS for wound healing. The paper includes physicochemical characterization, in vitro and in vivo studies. The in vitro and in vivo part are quite well described, however, the materials science part of the manuscript should be improved before publication.

Abstract: the role of HYP and Chitosan are not clear in the abstract, but it is present in the title, this results very confusing.

Introduction: The introduction should provide more information and references regarding the topic. The objective of the work is not clear, as mentioned before, the role of Chitosan is not properly described. More information summarizing the work must be added in the last paragraph of the introduction. Is the HYP included for release or as part of the structure?? To provide this information is essential to understand the potential of the manuscript.

Materials and methods: NMR experiments will give more information about the actual composition of the material and the amount of HYP and CS in the NPS as FTIR is not the optimal technique for this.

2.4 Characterizations of HYP-NPS nanoparticles techniques: must be briefly described.

Results: A table with the samples compared in vitro and in vivo including composition would be very useful.

Figures: must be aligned and present the same font size and style, please, correct that in all the figures.

Figure 1E the number can´t be seen

Figure 2A the Y axe shouldn´t have units and the must be label as arbitrary units X axe present a very weird scale, please, check that.

Figure 6. the significance of arrows and F, B and I must be explained in the figure caption.

Format details:

I recommend a thoroughly revision of spaces, for instance, between number and units and unified through the paper.  

Author Response

We are very grateful to the reviewer for his/her valuable comments and suggestions on our manuscripts. In the manuscript, we marked the revised parts in red. The answers to the questions of editors and reviewers are as follows:

Q1:The manuscript shows a complete work, which explore, for the first time the preparation of HYP- NPS for wound healing. The paper includes physicochemical characterization, in vitro and in vivo studies. The in vitro and in vivo part are quite well described, however, the materials science part of the manuscript should be improved before publication.

ANS:Thanks to the suggestions of the review experts, we supplemented the characterization method of the material part in the manuscript, accurately described the results and discussion part of the material part, and gave detailed operation steps and more meaningful discussion. The changes we made have been marked red in the manuscript.

Q2:Abstract: the role of HYP and Chitosan are not clear in the abstract, but it is present in the title, this results very confusing.

ANS:Thanks to the review experts for reminding us. We have modified the part of the abstract to increase the role of HYP and chitosan. Specific supplements are as follows:

Chronic wound repair is a common complication in patients with diabetes mellitus, which causes heavy burden on social medical resources and economy. Hypaphorine (HYP) has good anti-inflammatory effect. Chitosan (CS) is used in the treatment of wound because of its good antibacterial effect. The purpose of this research was designed to investigate the role and mechanism of HYP-nano microspheres in the treatment of diabetic rats wound healing. The morphology of HYP-NPS was watched by Transmission electron microscopy (TEM). RAW 264.7 macrophages were performed to assess the bio-compatibility of HYP-NPS. A full-thickness dermal wound in a diabetic rat model was performed to evaluate the wound healing function of HYP-NPS. The results revealed that HYP-NPS nanoparticles were spherical with an average diameter of about 50 nm. Cell experiments hinted that HYP-NPS had the potential as trauma material. The wound test in diabetic rats indicated that HYP-NPS fostered the healing of chronic wounds. The mechanism through down-regulating the expression of pro-inflammatory cytokines IL-1β and TNF-α in the wound skin, and accelerate the transition of chronic wound from inflammation to tissue regeneration. These results indicated that HYP-NPS had a good application prospect in the treatment of chronic wound.

Q3:Introduction: The introduction should provide more information and references regarding the topic. The objective of the work is not clear, as mentioned before, the role of Chitosan is not properly described. More information summarizing the work must be added in the last paragraph of the introduction. Is the HYP included for release or as part of the structure? To provide this information is essential to understand the potential of the manuscript.

ANS:Thanks to the suggestions of the review experts, we revised the background introduction in the manuscript, supplemented the disease statistics of chronic wound, the function description and research purpose of chitosan and HYP. Specific supplementary information has been marked in the manuscript.

Q4:Materials and methods: NMR experiments will give more information about the actual composition of the material and the amount of HYP and CS in the NPS as FTIR is not the optimal technique for this.

ANS:Thanks to the very practical suggestions provided by the review experts, we refer to a large number of research articles of the same type and find that FTIR spectroscopy is more suitable for studying the intermolecular binding mode, and NMR may be more suitable for studying the molecular structure of drugs. Therefore, here we choose FTIR spectroscopy to analyze the binding mode of CS and HYP. If necessary, we can do further research by NMR. Thank you again for your valuable suggestions.

Q5:2.4 Characterizations of HYP-NPS nanoparticles techniques: must be briefly described.

ANS:We briefly describe the characterization techniques of nanoparticles as follows: Briefly, the ZetaPALS was used to analyze the mean nanoparticles size and the zeta potential of 0.5 mg/mL HYP-NPS solution. The model of ZetaPALS was (Brookhaven Corporation). Each value was averaged from five parallel measurements. The rotary rheometer (DHR3; TA instruments, America) was used to measure the viscosity 5mg/mL HYP-NPS solution at room temperature 25℃ and body temperature 37℃, respectively. The shear rate was logarithmically distributed, ranging from 10-1s−1 to 1 to 102s−1. (n=3).FT-IR spectroscopy of HYP, NPS and HYP-NPS were taken with a potassium bromide precipitate on a spectrometer (Bruker Corporation, FT-IR TENSORâ…¡, Germany) with a wave number range of 500 to 4000 cm−1 at resolutions of 4 cm−1.

Q6:Results: A table with the samples compared in vitro and in vivo including composition would be very useful.

ANS:Thank the review experts for their suggestions. Considering that the drug HYP is a monomer, the content of HYP in the sustained-release solution was detected by HPLC in vitro. However, in vivo, it is difficult to extract HYP and CS from the wound site after they are absorbed by the wound site. In addition, HYP is also a small molecule drug. The metabolites at the wound site cannot be tracked because of their low content. Therefore, considering the above reasons, we did not list the sample composition.

Q7:Figures: must be aligned and present the same font size and style, please, correct that in all the figures.

ANS:Thank you for your reminding. We have modified all the pictures in the manuscript.

Q8:Figure 1E the number can´t be seen

ANS:Thanks to the review experts' reminder, we have modified the figure 1E.

Q9:Figure 2A the Y axe shouldn´t have units and the must be label as arbitrary units X axe present a very weird scale, please, check that.

ANS:Thanks to the review experts' reminder, we have revised figure 2A.

Q10:Figure 6. the significance of arrows and F, B and I must be explained in the figure caption.

ANS:Thank the review experts for reminding us. We have made descriptions of F, B and I in the figure legend to figure 6. I represent inflammatory cells, B represents blood vessels, and F represents fibroblasts.

Q11:Format details:I recommend a thoroughly revision of spaces, for instance, between number and units and unified through the paper.  

ANS:Thanks to the review experts' reminding, we have modified all the blanks in the manuscript and made unification.

Round 2

Reviewer 2 Report

The manuscript can be published as the authors have successfully improve it.